# Misdiagnosis and Mistreatment of a Rare Case of Intracranial Oncogenic Osteomalacia with an Altered Amino Acid Profile

**DOI:** 10.3390/medicina58121875

**Published:** 2022-12-19

**Authors:** Estella Musacchio, Alberto Michielin, Leonardo Sartori

**Affiliations:** 1Clinica Medica 1, Department of Medicine DIMED, University of Padova, 35128 Padova, Italy; 2Medicina Generale, Montebelluna General Hospital AULSS 2, 31044 Montebelluna, Italy

**Keywords:** amino acid profile, FGF23, muscle weakness, oncogenic osteomalacia, tumor-induced osteomalacia

## Abstract

*Background*. Oncogenic osteomalacia (OO), also known as tumor-induced osteomalacia (TIO), is a rare paraneoplastic syndrome caused by mesechymal tumors secreting fibroblast growth factor 23 (FGF23). Common in middle age, these tumors are often disclosed by progressive generalized bone pain and muscle weakness, along with an altered biochemical profile. Despite its characteristic presentation, the disease is often underrecognized with delayed onset of surgical or pharmacological intervention that can have serious repercussions on the patients’ health and quality of life. *Case presentation*. We describe the case of a 65-year-old Caucasian man presenting TIO with intracranial and spinal localizations and Fanconi-like aminoaciduria. The condition was misdiagnosed and mistreated for three years, leading to loss of self-sufficiency and depression. Following proper identification, the spinal mass was excised with complete remission of the functional symptoms. As it was not possible to remove the intracranial lesion, the patient was treated conservatively with calcitriol and phosphorous supplements that granted good metabolic control up to the time of a recent follow-up visit (at 5 years). *Conclusions*. The finding of an altered amino acid profile, not usually reported in these cases, should prompt clinicians to a wider usage of these molecules as suitable candidates for metabolic diseases. In addition to providing central information, they are easy to obtain and inexpensive to analyze. Such determination could help to speed up the diagnostic process, as a long-lasting history of misdiagnosis and mistreatments can lead primarily to clinical worsening, but also to the use of expensive, useless medications with side effects that contribute to poor patient health.

## 1. Introduction

Osteomalacia, a disease of adult age known as rickets in children, is a metabolic bone disease with impaired bone mineralization as a result of inadequate levels of available phosphate, calcium, and vitamin D. 

Oncogenic osteomalacia (OO) also known as tumor-induced osteomalacia (TIO) is a rare paraneoplastic syndrome caused by mesenchymal tumors secreting fibroblast growth factor 23 (FGF23). These tumors are small, mostly benign, slow-growing and difficult to locate [1,2,3]. The recurrence rate is low, but they are usually multiple. FGF23 is a key molecule in the mineral homeostasis. In physiological conditions, the osteocytes secrete dentin matrix protein (DMP1) and phosphate-regulating neutral endopeptidase (Phex) that downregulate FGF23 and induce renal and intestinal phosphate reabsorption, thus allowing regular bone mineralization. When FGF23 levels are high, as a consequence of low DMP1 and Phex or in cases of excess ectopic secretion, phosphate reabsorption is inhibited, its excretion is high, 1-alpha-hydroxylation of vitamin D is reduced and mineralization is impaired, causing rickets or osteomalacia.

Common in middle age, these tumors are often disclosed by the onset of progressive generalized bone pain and muscle weakness. Whole-body Tc-99m sestamibi scanning, octreotide scintigraphy, 18F-fluorodeoxyglucose (FDG) positron emission tomography (PET), whole body magnetic resonance imaging (MRI), and computed tomography (CT) are used for their localization [4,5].

Signs and symptoms can include diffuse body pain, muscle weakness, and fragility of the bones. In addition to low systemic levels of circulating mineral ions necessary for bone and tooth mineralization, accumulation of mineralization-inhibiting proteins and peptides, such as osteopontin and acidic serine aspartate-rich MEPE-associated motif (ASARM) peptides, occurs in the extracellular matrix of bones and teeth, likely contributing locally to causing matrix hypomineralization and osteomalacia [6].

Since the responsible tumor is usually benign, oncogenic osteomalacia has a good prognosis and surgical removal of the tumor results in a dramatic improvement [7].

We report the case of a TIO with intracranial and spinal localizations, misdiagnosed for a long time, with Fanconi-like aminoaciduria.

## 2. Case Presentation

A 65-year-old Caucasian man, complaining of increasing articular pain of the lower limbs over the preceding 3 yrs, initially at the right side, subsequently bilaterally, associated with myalgia and sporadic lumbar aching. Pain accompanied active and passive joint movements. Due to progressive muscle weakness and an impaired gait, the subject was confined to a wheelchair within a year.

Initial orthopaedic and rheumatological assessments revealed mild osteoporosis and spondyloarthritis with discal spine bulging in L3–L5, cervical osteophytosis, and initial wedging at the dorsal level. Electromyography (EMG), MRI and doppler ultrasound of lower limbs were negative for neurological or vascular involvement. During the same year, the patient underwent a variety of physical therapy strategies including water and massotherapy without any benefit. He then manifested initial depressive symptoms and was treated with alprazolam. 

Total body scintigraphy revealed hypercaptation of rib-costal joints with a marked rosary sign at the chest, and some other mildly captating regions (Figure 1).

One year later, an MRI investigation showed an initial osteonecrotic area in the right knee and trabecular metaphyseal fractures of the distal femur and proximal tibia (Figure 2A,B), as well as healed distal tibial and ankle stress fractures (not shown). 

In light of these findings, rheumatoid/inflammatory arthritis was diagnosed and the patient was consequently treated with disease-modifying antirheumatic drugs (DMARDs) and biological anti-TNF agents, but no improvement was reported. Notably, rheumatoid factor (RF), anti-nucleus antibodies (ANA), extractable nuclear antigens (ENAs), anticardiolipin (ACL), anti-mitochondrial antibodies (AMA), anti-DNA, anti-smooth muscle antibody (ASMA), liver kidney microsomal antibody (LKMA) and anti-Borrelia were all negative.

Three years later, the skeletal picture worsened with multiple vertebral fractures (not shown) and the patient was admitted to the local hospital for intensive physical therapy that proved to be unfeasible due to the severity of the symptoms. 

After a biochemical assessment that showed serum phosphates at 0.6 mg/dL (n.r. 2.3–4.7) and parathyroid hormone (PTH) levels at 93 ng/mL (12–72), the patient was transferred to the Central University Hospital for further investigation. 

The familial history revealed a 52-year-old brother with arthralgia of unknown origin. The physiological history disclosed delayed walking (2 yrs), initial rickets (likely treated with vitamin D) and subsequent normal development. The diet was varied and balanced, and included dairy products. The patient was a former heavy smoker (20 cigarette/day for 40 yrs). His current weight was 85 kg, with a significant decrease in the last 3 yrs.

The patient’s pathological history, which was not related to the current symptoms, included hypercholesterolemia, Wolf–Parkinson–White syndrome, surgery for cervical lipoma, and pneumonia. At the physical examination, the patient was quick and collaborative. The supine position was necessary because of limb pain. Nothing relevant was noted in the chest, heart, and abdominal exam. Muscular hypotrophy of the lower limbs and slow reflexes were noted.

Basic biochemistry testing disclosed only an elevated phosphorous excretion fraction. Second level biochemical analyses showed high FGF23 and high bone alkaline phosphatase (bALP) with type I collagen C terminal telopeptide (CTX) at the upper limit and low 1,25(OH)_2_vit D; all values are reported in Table 1. The investigation also included a urinary aminoacidic profile, which was highly altered, as reported in detail in Table 2.

Bone mass evaluation by dual energy X-ray absorptiometry (DEXA), reported in Figure 3, showed severe osteoporosis of both femurs (T-score −5.0 and −5.4 SD for left and right, respectively) and the lumbar spine (T-score −4.5 SD).

Although the DEXA cannot be considered a radiological examination/imaging diagnostic tool, the left femoral neck appeared much shorter, compatible with a closed fracture. Moreover, and more interestingly, at the T12/L1 spine level, an hyperdense area was clearly detectable. This was further investigated by the mean of Global PET Ga68, that showed hypercaptation at the expansive formation in T12-L1 with narrowing of the vertebral canal and bone lysis at the L1 vertebral body and transverse and spinous processes levels. Moreover, the exam denoted an increased uptake in a small encephalic area of the frontal cortex (Figure 4A,B). 

The patient was diagnosed with oncogenic osteomalacia, treated with phosphorous supplement (1000 mg/day), calcitriol 0.50 mcg/day and opioid painkillers, and thereafter transferred to a Surgical Unit. The timeline of the diagnostic process is schematized in Figure 5.

Follow up. The patient promptly underwent surgical decompression at the spine, followed by excision of the vertebral mass. The histological report confirmed the diagnosis of a mesenchymal phosphaturic tumor. 

Symptoms relief was fast and was paralleled by a good functional recovery. Within a few days, the patient was able to walk with a cane and for personal reasons did not accept brain surgery to remove the primary mass. The spinal surgery was further refined three years later.

The pharmacological therapy with calcitriol and P supplements is still ongoing. A recent follow-up, five years after the diagnosis, revealed extremely good overall conditions: physically, the patient does not complain about pain, he can walk for several kilometers without problems, and most of all he is in optimal psychological mood, considering himself as “someone who has had the privilege of living a second life”. 

The patient gave his informed consent to the publication of these data.

## 3. Discussion

We have described the case of a patient whose TIO diagnosis took 3 years to make. Although this is the average time elapsing from symptom onset to diagnosis, it must be stressed that nowadays diagnostic tools are many and sophisticated. Moreover, access to the scientific literature is wide and the combinations of these factors should prevent misdiagnosis and mismanaging. 

As soon as the patient was transferred to the University Hospital, he underwent a succession of tests that led to a diagnosis within two months. Some of these tests are quite sophisticated and expensive, such as the global PET that was, however, the last exam to confirm the presence of the tumor and localize the primary lesion. It must be pointed out that lab tests routinely made at hospital admission could point toward a diagnosis, or at least narrow the possibilities.

The finding of a high phosphorous renal fraction was suggestive of renal leaking. Renal causes leading to phosphaturic hypophosphatemia are extremely rare in adults. In the present case, they were excluded as the patient did not have any other specific symptoms, had a normal height, and was not organ transplanted. Among FGF 23-mediated causes, oncogenic osteomalacia was considered the most likely and therefore further investigated. 

The patient presented a severe and generalized Fanconi-like aminoaciduria that is not a frequent finding in TIO. The mechanism through which aminoaciduria is attained is extremely interesting and involves the vitamin D hormonal system. The main source of 1,25(OH)2vitD is the proximal tubule, the location of 1α-hydroxylase. Its gene CYP27B1 is regulated by a complex system mainly determined by FGF23 and phosphorous, but that also includes calcium, PTH, and the FGF23 co-receptor α-Klotho. FGF23 downregulates the sodium-phosphate transporter NaPi-2a in proximal tubule segments. This pathway is independent on FGF23-induced phosphaturia. 

As early as the 1950’s, it was found that low levels of vitamin D were associated with aminoaciduria [8], and several observations supported the possibility that such association was mediated by PTH, known to enhance phosphate excretion and cAMP production and thereafter calcium reabsorption. It was only in 1989 [9] that it was found that aminoaciduria in vitamin D deficiency is independent of PTH and urinary cAMP levels. It was discovered that the actual site of the pathway was located in the apical membrane, the site of active amino acid accumulation. More recent acquisitions in molecular biology disclosed that aminoaciduria in vitamin D deficiency relates to a failure of upregulation of amino acid transporter synthesis [10]. In the presence of high levels of FGF23, the consequent low vitamin D may be responsible for the development of aminoaciduria. The pathway is schematized in Figure 6 [11]. The patient’s 1,25(OH)2vitD levels were very low. We are not aware of the exact moment when this condition began, although we can hypothesize that it persisted for quite a long time, provoking diffuse aminoaciduria. 

The binding of FGF23 to its receptor FGFr occurs in the presence of the co-receptor klotho. Excess FGF23 provokes a fall in 1,25(OH)2vitD circulating levels. FGF23 also induces down-regulation of the NaPi transporter.

The 25OH vitD endocytic process is mediated by its binding to DBP. The internalization follows binding to the receptorial complex cubilin–megalin (cubilin, lacking the transmembrane domain needed for endocytosis, must colocalize with megalin). Inside the cell, 25OH vit D is the substrate for 1α-hydroxylase to synthetize the active metabolite 1,25(OH)_2_vit D. The enzyme activity is regulated by other factors, among which FGF23 and Pi are the major determinants, and PTH is also involved. 

Abbreviations: DBP: vitamin D binding protein; PTH: parathyroid hormone; Pi: phosphorus; FGFr: FGF receptor; VDR: vitamin D receptor; VDRE: vitamin D response element.

Myiagi et al. [12] measured plasma free amino acids of different cancer types stressing their great potential for improving cancer screening and diagnosis and understanding disease pathogenesis. We found an altered urine amino acid profile and we agree with these authors that amino acids are among the most suitable candidates for focused metabolomics as they are either ingested or synthesized endogenously and play essential physiological roles both as basic metabolites and metabolic regulators. Moreover, they are easily obtainable compared to other existing screening methods.

In conclusion, we want to point out that a long-lasting history of misdiagnosis and mistreatments results most of all in clinical worsening, but also in the use of expensive and useless medications with side effects that contribute to the patient’s overall bad conditions.

## Figures and Tables

**Figure 1 medicina-58-01875-f001:**
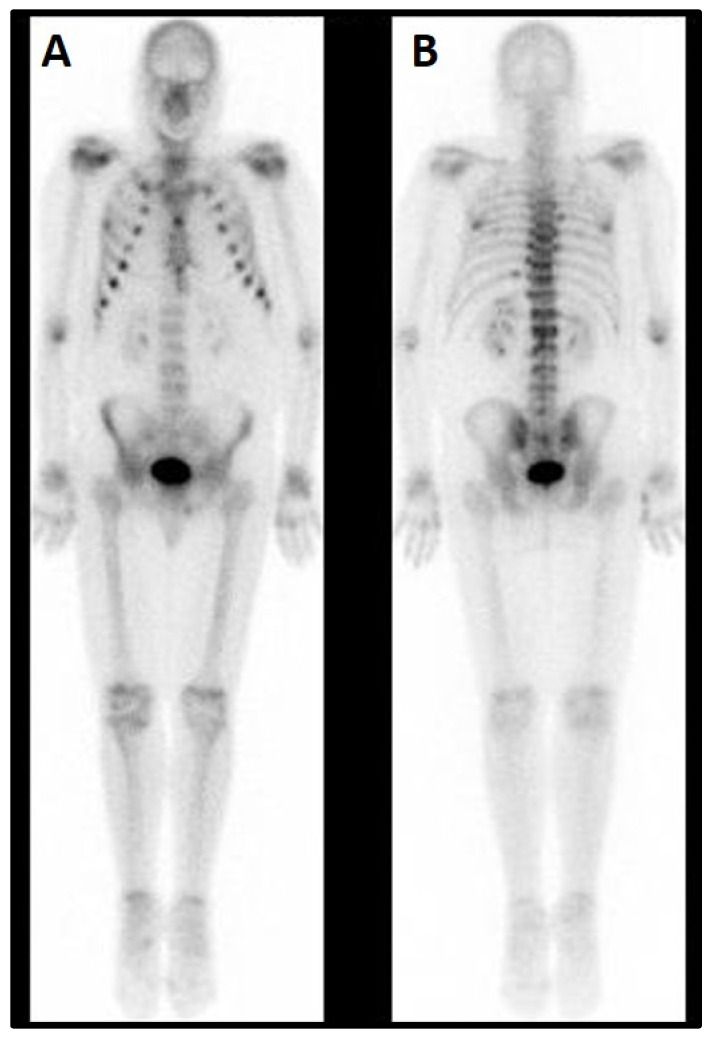
Total body scintigraphy (September 2015). (**A**) Front view. A Marked rosary sign at the chest. (**B**) Back view. Non-homogeneous spinal distribution of the tracer with increased uptake at the dorso-lumbar junction.

**Figure 2 medicina-58-01875-f002:**
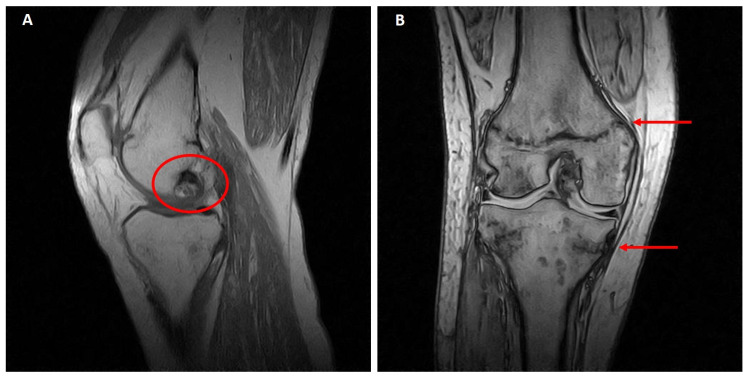
Right knee MRI (July 2016). (**A**) Osteonecrotic area of about 2 cm in the external femoral condyle. (**B**) The arrows indicate femoral and tibial metaphyseal fractures.

**Figure 3 medicina-58-01875-f003:**
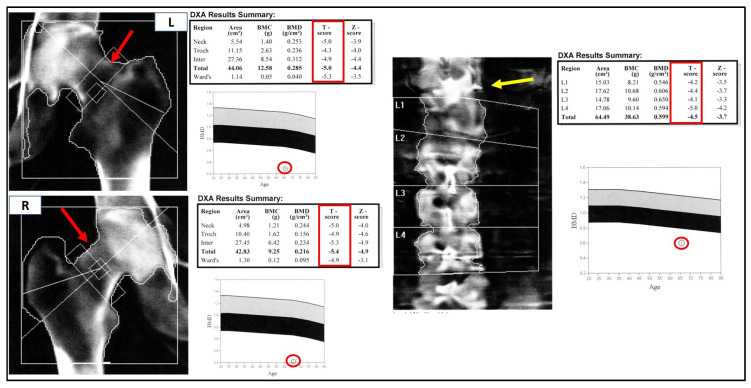
Bone mass evaluation by DEXA. The hips are shown at the right of the figure (left hip, L, above; right hip, R, below), the spine is on the left. The bone densitometry values (shown in the DXA results summary table) are far below the osteoporotic threshold (T-score < −2.5 SD). The inferior panels report the patient’s bone mineral density (BMD) levels compared to those expected at the same site for age-matched subjects: normal values above the grey band (osteopenic) and frankly osteoporotic below the black band (osteoporotic). The different length of the right and left femoral necks, indicated by the red arrows, may indicate a closed fracture. The hyperdense area at T12-L1 is indicated by a yellow arrow.

**Figure 4 medicina-58-01875-f004:**
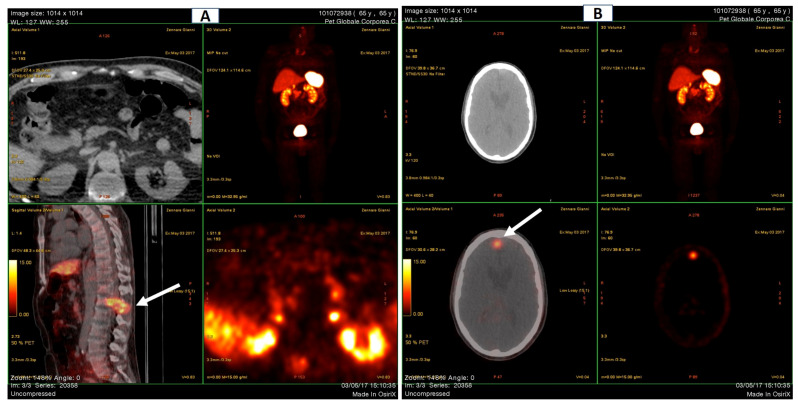
Global PET (Ga68-DOTATOC). (**A**) Spine. The white arrow indicates hypercaptation at T12/L1. (**B**) Skull. The white arrow indicates a focal area of increased uptake (max. axial diameter 13 mm).

**Figure 5 medicina-58-01875-f005:**
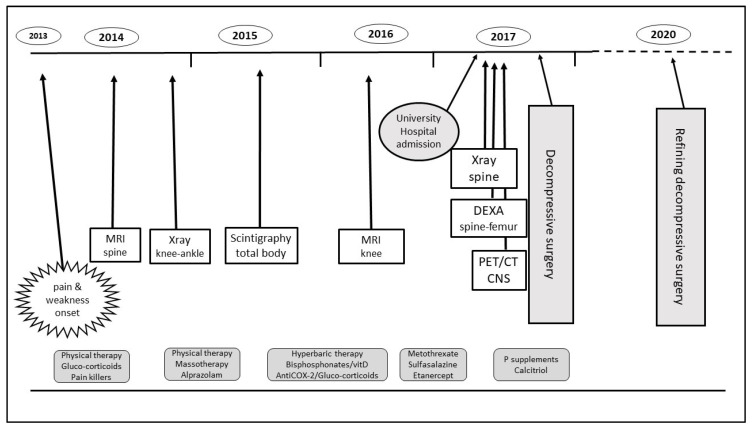
The timeline of Oncogenic osteomalacia (OO) diagnosis and patient management. CNS: central nervous system; DEXA: dual emission X-ray absorptiometry; MRI: magnetic resonance imaging; PET/CT: positron emission tomography/computed tomography.

**Figure 6 medicina-58-01875-f006:**
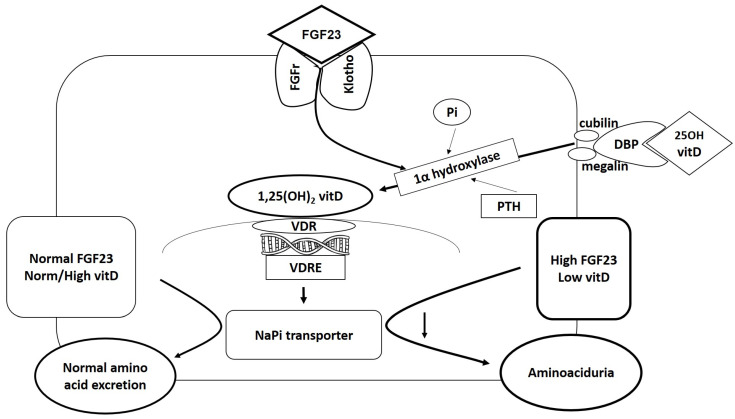
Pathway leading to aminoaciduria in the proximal tubule.

**Table 1 medicina-58-01875-t001:** Biochemical parameters at baseline.

	Value	Normal Range
**I level investigations**		
s-Creatinine (mg/dl)	0.58	
u-Creatinine (mg/dl)	72.32	
s-Ca (mmol/L)	2.29	2.10–2.55
u-Ca (mmol/L)	3.06	
s-P (mmol/L)	0.22	0.87–1.45
u-P (mmol/L)	12.64	12.9–42
PTH (ng/L)	37.7	4.6–26.8
**II level investigations**		
FGF 23 (pmol/L)	12.68	0–0.8
1,25(OH)2vitD (pmol/L)	29.8	62.6–228
bALP (µg/L)	85.5	3–20.2
CTX (pg/mL)	702.7	115–748
u-Glucose (mmol/24 h)	2.20	0–2.78

**Table 2 medicina-58-01875-t002:** Aminoaciduria (* indicates out of range values).

	Value µmol/24 h	Normal Range
1-Methyilhistidine	1076 *	68–855
2-Aminoadipic Acid	101	10–103
2-Aminobutirric Acid	111 *	9–45
Alanine	695	153–760
Arginine	19	13–64
Asparagine	608 *	25–202
Citrulline	30 *	0–29
Cystathionine	50 *	0–47
Cystine	160 *	28–115
Glutamic Acid	70 *	8–69
Glutamine	1801 *	111–822
Glycine	5465 *	501–2731
Histidine	2077 *	316–1665
Isoleucine	34	3–56
Leucine	72	12–84
Lysine	558 *	32–290
Methionine	22	8–73
Ornithine	81 *	5–70
Phenylalanine	621 *	16–175
Serine	1191 *	56–606
Taurine	1063	119–1661
Threonine	689 *	38–392
Tyrosine	275	18–288
Valine	85	8–85

## Data Availability

All data relevant to the study are reported in the manuscript.

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
