# Peer review of "Misdiagnosis and Mistreatment of a Rare Case of Intracranial Oncogenic Osteomalacia with an Altered Amino Acid Profile"

_medicina, 2022, doi:10.3390/medicina58121875_

Round 1

Reviewer 1 Report

The present manuscript describes the clinical course of undiagnosed oncogenic osteomalacia over the course of 3 years. The signs, symptoms, imaging findings, and laboratory testing results of this rare condition are thoroughly reviewed. The treatment was adequately described, with a purported full recovery 5 years later. No scientific inaccuracies are apparent in this brief report. The English language used in this article was sufficient to convey the main points, but it could benefit from language editing for improved clarity. Furthermore, the entire manuscript should be re-examined to ensure abbreviations and acronyms are described at first use.

Major comments:

1. Figure 1: This figure contains 2 panels; however, they are not labeled separately, and this is not acknowledged in the figure caption, which should describe each panel.

2. Figure 2: This figure contains 2 panels, which are presented here as two separate image files which must be combined into a single image. No figure caption is provided, only a figure title. As with Figure 1, each panel should be labeled A and B, and described individually in the figure caption.

3. Figure 3: As with Figure 2, the separate panels of Figure 3 must be combined into a single image file. Again, the figure caption is missing and must be provided, with details given for each panel.

4. In both Figure 3A and 3B, it appears potentially identifying information is present in the images under the heading “Scan Information”. In order to ensure that the data presented in this case report is properly de-identified, would recommend including only the anatomic images from this DEXA scan in the figure and placing the relevant medical findings in the figure caption.

5. As with other figures, Figure 4 must be presented as a single image, and the missing figure caption must be provided.

6. In addition to the figure title provided, Figure 5 is missing a figure caption as well. For examples of multipanel figures with full captions, previous case reports published in Medicina are available:

Lin, T.-Y.; Chien, K.-H. Delayed Onset Bilateral Papilledema in a Young Boy’s Eyes after Trauma. Medicina 2022, 58, 140. https://doi.org/10.3390/medicina58010140

Ha, J.-H.; Jeong, B.-H. Airway Foreign Body Mimicking an Endobronchial Tumor Presenting with Pneumothorax in an Adult: A Case Report. Medicina 2021, 57, 50. https://doi.org/10.3390/medicina57010050

Minor comments:

7. Figure 1: The abbreviation “TB” in not explained at first use in the main text nor is it explained when used in this figure title.

8. Figure 1: The space between the two panels is unnecessarily large.

9. Case presentation, ln 74: “FKT” is not explained at first use, nor does it appear to be an acronym commonly used in English. Would recommend using the relevant English expression instead. (Also: ln 101.)

Grammar:

1. In the title and throughout, “amino acid” should be written as two separate words.

2. Abstract, ln 17: “65-years-old” should be changed to “65-year-old”.

3. Introduction, ln 49: “Hydroxylation” is misspelled.

4. Introduction, ln 60: “And” is misspelled.

5. Case presentation, ln 67: “Complaining for” should be replaced with “complaining of”.

6. Case presentation, ln 97-99: According to journal guidelines: “When defined for the first time, the acronym/abbreviation/initialism should be added in parentheses after the written-out form.”

7. Case presentation, ln 140: “D12” should be “T12”.

Author Response

We would like to thank the Reviewers for the constructive comments that helped us to improve the quality of the paper and to apologize for the inaccuracies that were present in the previous version.

Our reply is reported in Italic below each reviewer's comment

As for the specific comments:

Reviewer 1

  1. Figure 1: This figure contains 2 panels; however, they are not labeled separately, and this is not acknowledged in the figure caption, which should describe each panel.
  2. Figure 1: The abbreviation “TB” in not explained at first use in the main text nor is it explained when used in this figure title.
  3. Figure 1: The space between the two panels is unnecessarily large.

Figure 1 has been modified according to the reviewer’s comments. The abbreviation TB has been replaced with “total body”.

  1. Figure 2: This figure contains 2 panels, which are presented here as two separate image files which must be combined into a single image. No figure caption is provided, only a figure title. As with Figure 1, each panel should be labeled A and B, and described individually in the figure caption.

Figure 2 has been combined as for the to the reviewer’s request. A descriptive caption has been added.

  1. Figure 3: As with Figure 2, the separate panels of Figure 3 must be combined into a single image file. Again, the figure caption is missing and must be provided, with details given for each panel.
  2. In both Figure 3A and 3B, it appears potentially identifying information is present in the images under the heading “Scan Information”. In order to ensure that the data presented in this case report is properly de-identified, would recommend including only the anatomic images from this DEXA scan in the figure and placing the relevant medical findings in the figure caption.

Figure 3 is now a single image and all the potentially identifying information was removed.

  1. As with other figures, Figure 4 must be presented as a single image, and the missing figure caption must be provided.

As well as the other figures, Figure 4 is also a single image with labelled panels and descriptive caption.

  1. In addition to the figure title provided, Figure 5 is missing a figure caption as well.

A caption with acronyms’ explanation has been added.

For examples of multipanel figures with full captions, previous case reports published in Medicina are available:

Lin, T.-Y.; Chien, K.-H. Delayed Onset Bilateral Papilledema in a Young Boy’s Eyes after Trauma. Medicina 2022, 58, 140. https://doi.org/10.3390/medicina58010140

Ha, J.-H.; Jeong, B.-H. Airway Foreign Body Mimicking an Endobronchial Tumor Presenting with Pneumothorax in an Adult: A Case Report. Medicina 2021, 57, 50. https://doi.org/10.3390/medicina57010050

We believe that our figures conform now to the ones of the cited articles.

Minor comments:

  1. Case presentation, ln 74: “FKT” is not explained at first use, nor does it appear to be an acronym commonly used in English. Would recommend using the relevant English expression instead. (Also: ln 101.)

We replaced “FKT” with “physical therapy” in both occurrences.

Grammar:

  1. In the title and throughout, “amino acid” should be written as two separate words.
  2. Abstract, ln 17: “65-years-old” should be changed to “65-year-old”.
  3. Introduction, ln 49: “Hydroxylation” is misspelled.
  4. Introduction, ln 60: “And” is misspelled.
  5. Case presentation, ln 67: “Complaining for” should be replaced with “complaining of”.
  6. Case presentation, ln 97-99: According to journal guidelines: “When defined for the first time, the acronym/abbreviation/initialism should be added in parentheses after the written-out form.”
  7. Case presentation, ln 140: “D12” should be “T12”.

We have made all the suggested changes and the manuscript has been checked by a native English-speaking expert.

Reviewer 2 Report

The authors report the case of a TIO with intracranial and s mistreatments results most of all in clinical worsening, but also in the use of expensive and pinal localizations, misdiagnosed for a long time, with Fanconi-like aminoaciduria. In conclusion the authors want to point out that a long-lasting history of misdiagnosis  and mistreatments results most of all in clinical worsening, but also in the use of expensive and useless medications with side effects that contribute to the patient’s overall bad conditions

The introduction is well written , with adequate bibliographic references.The clinical case is widely described with a very demonstrative iconography

The discussion is correct, adapting to the results obtained. It could be interesting to graphically represent the mechanisms that explain the interaction between vitamin D and aminoaciduria

Author Response

The discussion is correct, adapting to the results obtained. It could be interesting to graphically represent the mechanisms that explain the interaction between vitamin D and aminoaciduria.

We tried to schematize the possible pathway leading to aminoaciduria providing a figure with an explicative caption (Figure 6 in black and white while the same in colours is in the supplemental files). A sentence and a new reference (#11) were added in the discussion.  

Round 2

Reviewer 1 Report

The present version of the manuscript represents a significant improvement and addresses all the previous concerns. No additional changes are necessary from my perspective.